# SEVA: Leveraging sketches to evaluate alignment between human and machine visual abstraction

Kushin Mukherjee[1],[*] Holly Huey[2],[*] Xuanchen Lu[2],[*] Yael Vinker[3], Rio Aguina-Kang[2], Ariel Shamir[4], and Judith E. Fan[2],[5]

University of Wisconsin-Madison[1]
University of California, San Diego[2]
Tel-Aviv University[3]
Reichman University[4]
Stanford University[5]

## Abstract

Sketching is a powerful tool for creating abstract images that are sparse but meaningful. Sketch understanding poses fundamental challenges for general-purpose vision algorithms because it requires robustness to the sparsity of sketches relative to natural visual inputs and because it demands tolerance for semantic ambiguity, as sketches can reliably evoke multiple meanings. While current vision algorithms have achieved high performance on a variety of visual tasks, it remains unclear to what extent they understand sketches in a human-like way. Here we introduce SEVA, a new benchmark dataset containing approximately 90K human-generated sketches of 128 object concepts produced under different time constraints, and thus systematically varying in sparsity. We evaluated a suite of state-of-the-art vision algorithms on their ability to correctly identify the target concept depicted in these sketches and to generate responses that are strongly aligned with human response patterns on the same sketch recognition task. We found that vision algorithms that better predicted human sketch recognition performance also better approximated human uncertainty about sketch meaning, but there remains a sizable gap between model and human response patterns. To explore the potential of models that emulate human visual abstraction in generative tasks, we conducted further evaluations of a recently developed sketch generation algorithm [91] capable of generating sketches that vary in sparsity. We hope that public release of this dataset and evaluation protocol will catalyze progress towards algorithms with enhanced capacities for human-like visual abstraction.

## 1 Introduction

Abstraction is key to how humans understand the external world. Abstraction enables distillation of individual sensory experiences into compact latent representations that support learning of new concepts [80, 45, 66, 30] and efficient communication about these concepts with others [23, 85, 34, 28]. For example, while no two roses are identical, people can rapidly infer what properties make a flower a *rose* and not some other kind of flower from just a few examples [99, 50], especially when these examples are selected to support such strong inferences [32, 77].

---

[1]/[*]Equal contribution

37th Conference on Neural Information Processing Systems (NeurIPS 2023) Track on Datasets and Benchmarks.

## 1.1 Human Visual Abstraction as Key Target for AI

*Visual abstraction* enables humans to express what they know about the visual world by creating external representations that highlight the information they judge to be most relevant in any given context—for instance, pictures that highlight the visual features that are diagnostic of a *rose* in a botanical field guide [24, 23, 92]. Critically, there are many different ways to depict even the same object—from a detailed illustration to a simple sketch. The Spanish artist Pablo Picasso famously demonstrated this point in *The Bull* (1945), a series of 11 lithographs of bulls, each sparser than the last (Fig. 1). While some of the drawings in this series look more realistic and others more stylized, all of these images remain evocative of a *bull* (and perhaps other similar animals, such as a *moose* or *buffalo*) to most human viewers.

Drawing is one of the most accessible, enduring, and versatile techniques that humans in many cultures use to encode ideas and emotions in visual form [41, 1, 31]. Even without special training, humans can robustly produce and understand simple line drawings or sketches of familiar visual concepts [24, 81, 43]. The ability to leverage drawings to understand and convey key aspects of the visual world emerges early in childhood [62, 5, 44] and improves throughout development with children's expanding conceptual knowledge [19, 59, 42]. Moreover, failures to produce and recognize drawings of objects are associated with semantic dementia [10, 73], suggesting links between a robust capacity for visual abstraction and the organization of semantic knowledge in the brain.

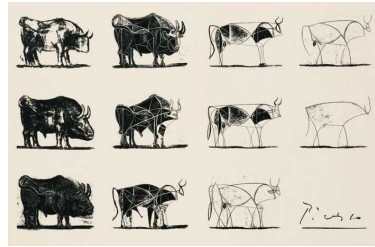

Figure 1: Pablo Picasso. *The Bull*, 1945.

In addition to drawings that represent objects and scenes, other abstract human-made visualizations (e.g., maps, diagrams, charts, graphs) serve important functions in many domains, including all branches of science and engineering [89, 90, 39, 13, 11]. Given the ubiquity and importance of such visualizations in modern life, developing computational models that achieve human-like understanding of freehand sketches is an important milestone. Such computational models of *human* visual abstraction stand to not only advance our understanding of human intelligence, but to also make AI systems more robust and general [40, 63, 27]. For example, prior work has found that incorporating principles based on the structure and function of the human visual system have led to vision models that are more robust (e.g., to adversarial attacks) [6, 55, 25].

## 1.2 Desiderata for Evaluating Alignment Between Human and Machine Visual Abstraction

The past several years have seen remarkable progress in the development of increasingly performant general-purpose vision algorithms [78, 36, 20, 70], with some of the most prominent algorithms also capable of emulating key aspects of how the primate brain encodes natural visual inputs [102, 48, 110, 47]. Over the same period, artificial vision systems have also been steadily achieving higher performance on tasks involving abstract visual inputs, including sketch categorization [21, 108, 4, 106], sketch segmentation [54, 104], sketch-based image/shape retrieval [22, 74, 105, 82, 9], among others [88, 100, 60, 16]. Moreover, these models have been found to predict human behavior on sketch recognition tasks to some degree [24, 23]. However, these otherwise high-performing vision algorithms struggle to simultaneously achieve robust understanding of visual inputs across multiple levels of abstraction [3, 79, 23]. Moreover, one study found that current vision models trained on natural images still fall short of the representational capabilities of the inferotemporal cortex, a key brain region supporting object categorization, in generalizing to new image distributions, including sketches [2]. Nevertheless, more recently developed models trained on substantially larger and varied datasets show promise on both tasks involving images with different visual styles [70, 107, 76] and tasks that go beyond recognition, including sketch generation [91, 69, 97] and sketch-guided image generation [61, 109, 96, 93, 56].

At present, it remains unclear to what degree any state-of-the-art models achieve *human-like* understanding of line drawings that vary in their degree of abstraction, much less the full range of abstract images that humans regularly engage with. Gaining further clarity on this question requires meeting two key challenges: *first*, creating a dataset containing drawings of a wide variety of object concepts that also systematically vary in their degree of abstraction; and *second*, developing evaluation protocols that can be used to estimate the degree to which any model emulates *human-like* understanding of this suite of drawn images.

**Dataset.** Meeting the first challenge requires going beyond existing sketch datasets [21, 74, 43, 18, 22, 54, 105, 82, 26, 95, 75, 51–53, 67, 111, 57]. While most of these datasets span a reasonably wide range of visual concepts (i.e., ranging from 125 concepts in *Sketchy* to 345 in *Quickdraw*) and some of them contain fine-grained information (e.g., stroke information and photo-sketch pairing), none of them systematically varied how detailed individual sketches could be, one of the most straightforward ways of inducing variation in semantic abstraction [8, 23, 103]. Our paper addresses the gap by providing sketches with controlled levels of detail, while encompassing the variety and granularity present in existing datasets (Table 1).

**Evaluation protocol.** Meeting the second challenge requires going beyond simple accuracy-based model performance metrics alone. Instead, it is critical to measure detailed patterns of human behavior on the same sketch understanding tasks to evaluate how well any model emulates these behavioral *patterns*, following recent work in the computational neuroscience of vision [71, 7, 68].

### 1.3 SEVA: A Novel Sketch Benchmark for Evaluating Visual Abstraction in Humans and Machines

In recognition of the above desiderata, here we introduce SEVA (**S**ketch-based **E**valuations of **V**isual **A**bstraction), a new sketch dataset and benchmark for evaluating alignment between human and machine visual abstraction.

**Dataset.** Our dataset contains approximately 90K human-generated sketches of a wide variety of visual objects that also systematically vary in their level of detail, and thus the variety of meanings they evoke. Each sketch is associated with one of 2,048 object instances belonging to one of 128 object categories selected from the THINGS dataset [38]. We achieved variation in sketch detail by imposing constraints on how much time humans ($N$=5,563 participants) had to produce each sketch (i.e., 4s, 8s, 16s, 32s).

**Evaluation protocol.** Leveraging these human-generated sketches, we systematically evaluated how well a diverse suite of 17 state-of-the-art vision models generate classification responses that align with those produced by humans ($N$=1,709 participants) tasked with identifying the most appropriate concept label for each sketch. Of these participants, 579 participants also participated in the sketch production study but were not shown any of their own sketches during this study. We evaluated human-model alignment using three different metrics: (1) top-1 classification accuracy, reflecting raw sketch recognition performance; (2) Shannon entropy of the response distribution, reflecting the degree of uncertainty about the target label; and (3) *semantic neighbor preference*, reflecting the degree to which models and humans generated off-target responses that were semantically related to the target label.

**Summary of key findings.** We found that sparser human sketches produced under more severe time pressure (e.g., 4 seconds) exhibited greater *semantic ambiguity*—in other words, both humans and models assigned a greater variety of labels to them than to the more detailed sketches that took more time to make (e.g., 32 seconds). Furthermore, we found that models that better predicted human sketch recognition performance also better approximated human uncertainty about sketch meaning, but none of the models came close to approximating human response patterns to human-generated sketches at any level of detail. To explore the potential of models that emulate human visual abstraction in generative tasks, we conducted further evaluations of CLIPasso, a recently developed sketch generation algorithm [91] capable of generating sketches that vary in sparsity (measured by number of constituent strokes in a sketch). We discovered that the most detailed CLIPasso-generated sketches converged with human sketches of the same object concepts, as measured by the distribution of labels humans assigned to sketches made by both agent types; however, sparser CLIPasso sketches diverged from human sketches of the same object concepts, reflecting a gap between how CLIPasso and human participants attempted to preserve sketch meaning under more severe production constraints.

## 2 Methods

### 2.1 Human Sketch Production

A core contribution of this work is a new dataset containing human-generated sketches of a wide range of visual object concepts that also systematically span multiple levels of semantic abstraction. We created this dataset by crowdsourcing these sketches online, following prior work [24, 105, 74, 23, 34, 43]. Each sketch in the dataset was recorded as a bitmap image as well as a collection of stroke coordinates, thus preserving the precise cursor movements a participant enacted to create the sketch.

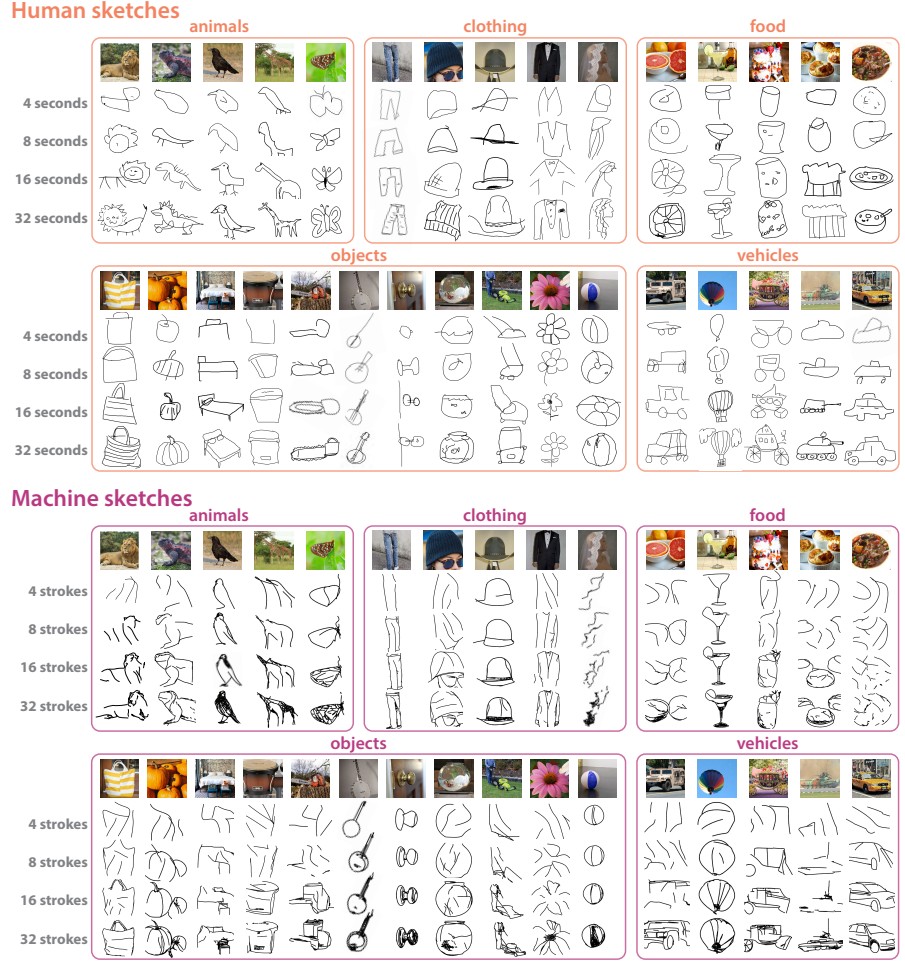

Figure 2: Humans and `CLIPasso` generated approximately 90K sketches under various production constraints.

**Participants.** 5,563 participants (2,870 male; $M_{age}$ = 36.7 years) were recruited from Prolific and compensated $15.50/hour for their participation. Data from 104 of these sessions were excluded from subsequent analyses due to technical issues (e.g., images did not load). All participants provided informed consent in accordance with the UC San Diego IRB.

**Object concepts.** We included 128 concrete real-world object categories (e.g., "lion", "banjo", "car") sourced from the `THINGS` dataset. We used the `THINGS` dataset [38, 37] because it is a well validated set of concrete, real-world visual object categories designed to support interoperability among large-scale studies in human visual cognition and cognitive neuroscience. For each of these 128 object concepts, we randomly sampled 16 object instances represented by color photographs, which served to visually ground the human sketch production task. As such, each sketch in our dataset is uniquely associated with one of these 2,048 object instances, and our final sample size was determined by our predefined goal of obtaining at least 10 human sketches of each of these instances.

**Sketch production task.** In each session, participants produced sketches of 16 different object categories, randomly sampled from the full set of 128 object categories. On each trial, they were cued with a color photograph (500px × 500px) of an object paired with its concept label. Each participant was randomly assigned to one of four conditions, defined by the maximum amount of time participants could take to produce their sketches: 4 seconds, 8 seconds, 16 seconds, or 32 seconds (Fig. 2, *left*). Such random assignment of participants to condition ensures that estimates of differences between conditions will not, in expectation, be biased by individual differences in sketching behavior. Participants drew on a digital drawing canvas (500px × 500px) using whatever input device they already had available (e.g., mouse, stylus) and were able to undo their most recent

| Dataset Name | Dataset Contents | # Classes | Stroke Info? | Photo Cue? | Abstraction? |
|---|---|---|---|---|---|
| TU-Berlin [21] | 20K sketches | 250 | ✓ | | |
| QuickDraw [43] | 50M sketches | 345 | ✓ | | |
| QuickDrawExtended [18] | 330K sketches, 204K photos | 110 | | | |
| SPG [54] | 20K sketches w/ stroke grouping | 25 | ✓ | | |
| SBSR [22] | 1.8K sketches, 1.8K 3D models | 161 | | | |
| QMUL [105, 82] | 1.3K sketches, 1.3K photos | 3 | | ✓ | |
| Sketchy [74] | 75K sketches, 12K photos | 125 | ✓ | ✓ | |
| SketchyCOCO [26] | 14K sketches, 14K photos | 17 | ✓ | ✓ | |
| **SEVA** | **90K sketches, 2048 photos** | **128** | ✓ | ✓ | ✓ |

Table 1: Comparison between SEVA and prior sketch datasets.

stroke or completely clear their canvas if needed. They were encouraged to make their drawings as recognizable as they could at the concept level and to use the photograph only to remind them of what individual objects belonging to that category generally look like. A countdown timer indicated how many seconds they had left to produce their drawing. Each trial ended either when time ran out or when the participant indicated that they wished to continue to the next trial, but participants were encouraged to use the full time available to produce as recognizable of a drawing as they could. At the beginning of the session, participants were explicitly instructed not to include any background context (e.g., grass in a drawing of a "horse"), arrows, or text. Participants also completed one practice trial (that we did not include in analyses) at the beginning of the session to familiarize themselves with the drawing interface. Our final dataset contains 89,797 sketches after filtering out invalid sketches (e.g., blank canvases).

## 2.2 Human Sketch Understanding

A key component of any evaluation of how well current vision algorithms emulate human visual abstraction is measurement of human behavior in tasks relying on visual abstraction. Here we focus on characterizing what meanings humans extract from the collected sketches, providing the basis for our subsequent empirical evaluation of how well any state-of-the-art vision model approximates human response patterns when presented with the same sketches.

**Participants.** 1,709 participants (776 male; $M_{age}$ = 39.2 years) were recruited from Prolific and compensated \$15.50/hour for their participation. Data from 21 of these sessions were excluded from subsequent analyses due to technical issues. Our predefined criterion for stopping data collection was acquisition of at least 12 recognition judgments for each sketch.

**Sketch recognition task.** In each session, participants provided labels for 64 sketches randomly sampled from a fixed set of 8,192 sketches, approximately 10% of the full human sketch dataset. The specific set of 8,192 sketches included in this experiment was determined by randomly sampling one sketch cued by each object instance from each drawing-time condition (i.e., 16 instances/concept × 128 categories × 4 drawing-time conditions = 8,192). On each trial, participants were presented with a single sketch (300px × 300px) and a text field where they could provide their best guess concerning the concept the sketch was intended to convey. As soon as they began typing, a drop-down menu appeared with suggested word completions. This drop-down menu contained the entire set of 1,854 labels in the THINGS dataset and only responses that matched one of these 1,854 labels were accepted. Because many words have multiple meanings, the labels contained in this dropdown menu were also accompanied by disambiguating text (e.g., to distinguish *mouse (animal)* from *mouse (computer)*). If participants were unsure of which label best applied to the sketch, they were encouraged to provide additional guesses (up to 5 per sketch). Collecting multiple labels on each drawing trial was important because it enabled us to more thoroughly sample the distribution of meanings that each sketch evoked for human participants (i.e., which labels came to mind and how often they did so). At the beginning of the session, participants completed a practice trial (that we did not include in analyses) to familiarize themselves with the labeling interface.

## 2.3 Machine Sketch Understanding

We propose a generic protocol for evaluating machine sketch understanding that can be applied to any vision algorithm using our sketch dataset. In this paper, we conduct evaluations of a wide range of state-of-the-art vision models with the goal of demonstrating the feasibility of our protocol and guiding future model development.

**Model Suite.** Specifically, we evaluated 17 vision models spanning a wide range of architectures and training methods (Table 2), all of which have been demonstrated to achieve high performance on object recognition on standard datasets, such as ImageNet [17]. We also made sure to include variants of standard ConvNet and Transformer models that have gained traction within the field of computational cognitive neuroscience for their potential to close the gap between biological and artificial vision [46, 25, 49, 65].

| Model | Architecture | Training Paradigm | Dataset |
|---|---|---|---|
| VGG-19 [78] | VGG-19 | supervised | ImageNet |
| Inception-V3 [84] | Inception-V3 | supervised | ImageNet |
| ResNet-50 [36] | ResNet-50 | supervised | ImageNet |
| ViT-B [20] | ViT-B | supervised | ImageNet |
| Swin-B [58] | Swin-B | supervised | ImageNet |
| MLPMixer-B [87] | MLPMixer-B | supervised | ImageNet |
| CORnet-S [49] | CORnet-S | supervised | ImageNet |
| Harmonization [25] | ViT-B | supervised | ImageNet + Human Feature Importance [25] |
| ECOSET [65] | ResNet-50 | supervised | ECOSET [65] |
| SimCLR [14] | ResNet-50 | self-supervised | ImageNet |
| MoCo-v3 [15] | ViT-B | self-supervised | ImageNet |
| DINO [12] | ViT-B | self-supervised | ImageNet |
| MAE [35] | ViT-B | self-supervised | ImageNet |
| CLIP [70] | ViT-B | self-supervised | WebImageText [70] |
| IPCL [46] | AlexNet | self-supervised | ImageNet |
| Noisy Student [98] | EfficientNet-b4 | semi-supervised | ImageNet + JFT [83] |
| SWSL [101] | ResNet-50 | semi-supervised | ImageNet + YFCC-100M [86] + IG-1B-Targeted [101] |

Table 2: Model suite annotated by backbone architecture, training paradigm, and training dataset.

**Evaluation Protocol.** The goal of our evaluation protocol was to measure how well any of these vision models approximated human sketch recognition behavior when presented with the same sketches. Because these models contain different latent representations of widely varying dimensionalities, we measured machine sketch-recognition behavior by extracting activation patterns from each model's final convolutional or attention block and training linear classification-based readouts on these activation patterns. That is, for each model we independently fit 1,854-way logistic regression classifiers using 5-fold stratified cross-validation to predict the "ground-truth" concept label associated with each sketch. [1]

These predicted labels were aggregated across the 16 sketches of the same concept (e.g., *lion*) and from the same drawing-time condition (e.g., 4 seconds) to yield a model's response distribution for that *type* of sketch. The top-1 classification accuracy was determined by computing the relative frequency of the "ground-truth" concept label in this response distribution. We also computed the Shannon entropy of this response distribution to estimate the degree of semantic ambiguity exhibited by this type of sketch. Further, we derived a measure of the degree to which even the *non*-ground-truth labels generated by each model were semantically related to the ground-truth label, which we term the *semantic neighbor preference* score. This semantic neighbor preference score falls in the range $[0, 1]$ and is highest when labels that are more semantically related to the ground-truth label appear more frequently than more semantically distant labels, is close to 0.5 when labels appear with uniform probability, and is minimized when labels that are more semantically distant appear most frequently.

To compare model classification outputs with human responses, we derived an analogous response distribution from the human labels obtained in the human sketch recognition experiment. That is, we aggregated all labels assigned by human participants to all 16 sketches of the same concept (e.g., *lion*) and from the same drawing-time condition (e.g., 4 seconds) to construct a response distribution for each type of sketch. We then computed the same three metrics above (i.e., top-1 classification accuracy, response entropy, semantic neighbor preference) using the human response distributions.

---

[1]Because these classification-based readouts were only trained on sketches of 128 object concepts subsetted from the THINGS dataset, the probabilities assigned to the remaining 1,726 labels were set to zero during training. As such, none of these models produced any of these other labels at test time, but humans in the sketch recognition experiment *could* select these other labels. This difference between how sketch recognition behavior was elicited from humans and models led us to focus on relative measures of performance when evaluating human-model alignment.

| time | accuracy$_{mean}$ | accuracy$_{sem}$ | entropy$_{mean}$ | entropy$_{sem}$ | SNP$_{mean}$ | SNP$_{sem}$ |
|---|---|---|---|---|---|---|
| **4 seconds** | .031 | .003 | 1.958 | .011 | .628 | .008 |
| **8 seconds** | .082 | .004 | 1.814 | .013 | .701 | .009 |
| **16 seconds** | .139 | .006 | 1.690 | .014 | .750 | .010 |
| **32 seconds** | .199 | .007 | 1.555 | .015 | .787 | .010 |

Table 3: Human sketch understanding under each draw duration constraint. Columns represent means and standard errors of the mean for top-1 accuracy, response entropy, and semantic neighbour preference (SNP).

## 2.4 Machine Sketch Production

To explore the potential of models that emulate human visual abstraction in generative tasks, we also include evaluations of CLIPasso, a recently developed sketch generation algorithm [91] capable of generating sketches that vary in sparsity.

**Generating machine sketches.** Specifically, we leveraged CLIPasso to generate 8,192 sketches conditioned on the same 2,048 object instances we used in the human sketch production experiment such that each sketch was constrained to consist of either 4, 8, 16, or 32 pen strokes (Fig. 2, *right*). CLIPasso generates sketches by optimizing the parameters of a set of curves (i.e., start/end points; control points), each representing a single pen stroke, to be similar to target image. This optimization is guided by a pretrained implementation of CLIP [70], a large model trained using contrastive learning on vast quantities of text-image pairs. Similarity to the target image is defined based on the distance between CLIP's embedding of the target image and its embedding of the sketch, where these embeddings reflect combinations of feature activations from multiple intermediate layers of CLIP.

**Measuring human understanding of machine sketches.** We evaluated human recognition performance on these CLIPasso-generated sketches by recruiting 1,481 participants (730 male; $M_{age}$ = 41.05 years) on Prolific to complete the same sketch recognition task described earlier. Data from 7 of these sessions were excluded from subsequent analyses due to technical issues.

## 3 Results

**Humans produce sparser sketches under stronger time constraints.** We first sought to validate the effect of manipulating the maximum time that human participants had to draw on how detailed their sketches were. We estimated how detailed a sketch was by counting the number of strokes it contained (Fig. 3) and then fit a mixed-effects linear regression model predicting the number of strokes as a function of drawing-time condition (i.e., 4s, 8s, 16s, 32s), with random intercepts for object concept. We found that drawings produced under the 4s limit contained the fewest strokes on average, whereas those produced under the 32s limit contained the greatest number of strokes ($\beta = .29$, $SE = 4.95 \times 10^{-3}$, $p < .001$). These results confirm that restricting the amount of time human participants had to produce their sketches led to systematic differences in how detailed their sketches were.

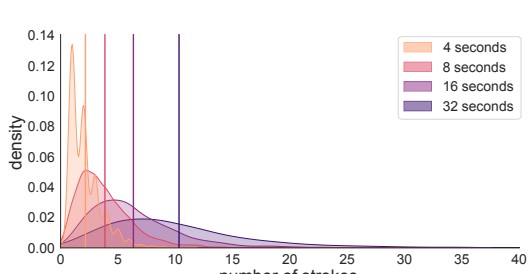

Figure 3: Distributions of number of strokes for each drawing time condition in the human sketch production task. Vertical lines indicate means.

**Sparser sketches are more semantically ambiguous for models and humans.** Having verified that we had successfully manipulated the level of detail in human sketches, we next sought to evaluate how well current vision models extract semantic information from them at each level of detail. Our general approach was to fit mixed-effect linear regression models to estimate the effect of sparsity on 3 key metrics: (1) top-1 classification accuracy, (2) entropy of response distributions, and (3) semantic neighbor preference. Regression models included random intercepts and slopes for object concepts. Figure 4 shows the performance of each vision model with respect to these metrics for sketches produced under different time constraints.

We found that models generally achieved higher top-1 classification accuracy for more detailed sketches than sparser ones ($\beta = 9.57 \times 10^{-2}$, $t = 20.25$, $p < .001$).

We further found the entropy of the models' response distribution was lower for detailed sketches than for sparser sketches ($\beta = -.03$, $t = -30.19$, $p < .001$), suggesting greater uncertainty about the best label to apply to sparser sketches. Even when sketches were more ambiguous, however, models generated labels that were semantically related to the ground-truth label, as measured by our *semantic neighbor preference* score, with more detailed sketches eliciting a greater proportion of semantically related labels ($\beta = 2.5 \times 10^{-2}$, $t = 8.43$, $p < .001$). These patterns were mirrored in human sketch recognition behavior, with more detailed sketches being associated with higher top-1 classification performance ($\beta = 6.08 \times 10^{-2}$, $t = 9.57$, $p < .001$), a tighter distribution of responses (lower entropy) ($\beta = -.14$, $t = -12.29$, $p < .001$), and greater semantic neighbor preference ($\beta = 5.41 \times 10^{-2}$, $t = 20.24$, $p < .001$).

**Different models display distinct patterns of sketch recognition behavior.** Although all vision models were sensitive to the effect of our drawing-time manipulation, we found that there were reliable differences in classification accuracy between models ($\chi^2(16) = 3455.3$, $p < .001$). Moreover, some models generated a greater diversity of responses than others, as measured by the entropy of their response distribution (Fig. 4B, $\chi^2(16) = 89698$, $p < .001$). Finally, models varied in the degree to which they generated non-ground-truth labels that were semantically related to the ground-truth label ($\chi^2(16) = 318.46$, $p < .001$). Taken together, these results indicate that these models, all high-performing, display systematic differences in how they extract semantic information from sketches.

**A large gap remains between human and model sketch understanding.** While both humans and models are affected by the amount of detail in sketches, it is not yet clear to what

Figure 4: Effect of drawing time constraints on sketch understanding in different vision models.

degree their response patterns are well aligned. We evaluated human-model alignment scores using the same three metrics (i.e., top-1 classification accuracy, entropy, semantic neighbor preference) by estimating the degree to which model performance on different *types* of sketches (e.g., lions drawn in 4 seconds or less) covaried systematically with human performance on the same *types* of sketches. For example, a model is considered well aligned with humans with respect to recognition performance if it achieves high top-1 classification accuracy on the same types of sketches that humans succeed in classifying *and* if it achieves low accuracy on the types of sketches that humans fail to classify. Similarly, a model is considered well aligned with humans with respect to semantic ambiguity if it produces a response distribution with high entropy for the same types of sketches that humans are also highly uncertain about *and* if it produces a low-entropy response distribution for the types of sketches that humans systematically agree on (regardless of whether this agreement is concentrated on the correct label). We found that models generaly displayed some degree of alignment to humans on top-1 classification accuracy ($\beta = 7.70 \times 10^{-2}$, $t = 6.77$, $p < .001$), response entropy ($\beta = 1.17 \times 10^{-1}$, $t = 28.52$, $p < .001$), and semantic neighbor preference ($\beta = 6.66 \times 10^{-2}$, $t = 6.21$, $p < .001$). Moreover, we found that different vision models aligned with humans to varying degrees (top-1 classification accuracy: $\chi^2(16) = 134.81$, $p < .001$; response entropy: $\chi^2(16) = 725.78$, $p < .001$; semantic neighbor preference: $\chi^2(16) = 5.05$, $p = .99$). Nevertheless, a sizable gap remains between the most aligned models and a human-human consistency baseline for *all* metrics (Fig 5; top-1 classification accuracy: $t = 952.19$, $p < .001$; response entropy: $t = 184.21$, $p < .001$; semantic neighbor preference: $t = 389.56$, $p < .001$). Finally, we observe that while examining classification accuracy and entropy yield similar rankings over which models are best aligned to humans, these metrics appear to capture non-redundant sources of information about human and model sketch understanding (Fig 5D).

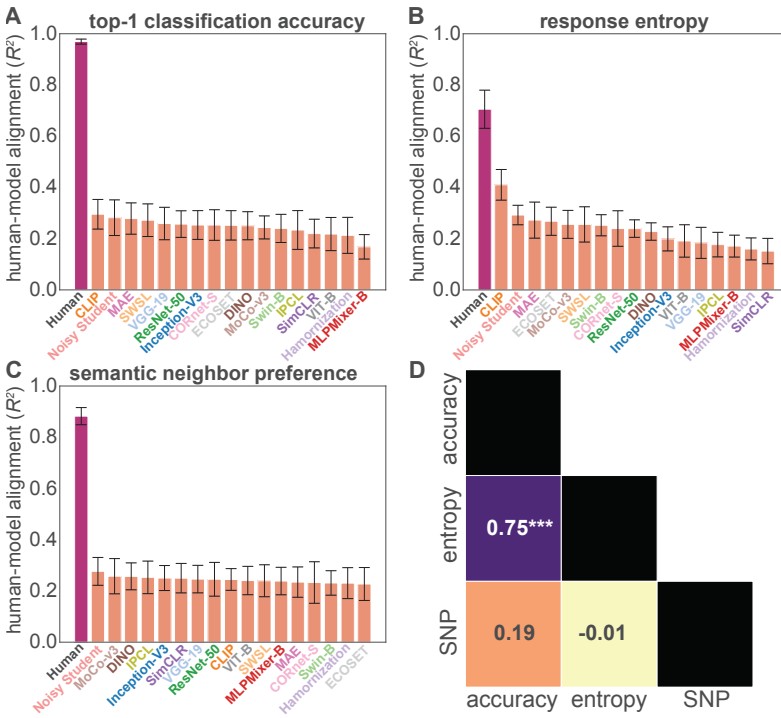

Figure 5: Human-model alignment on (A) top-1 classification accuracy, (B) response entropy, and (C) semantic neighbour preference. Leftmost red bars in each plot correspond to baseline human-human consistency on each metric. Error bars indicate bootstrapped 95% confidence intervals. (D) Spearman $\rho$ correlations between the rank-ordering of vision models with respect to their alignment to human performance on each metric.

**A CLIP-based sketch generation algorithm emulates human sketches under some conditions.**

While sketch understanding is a critical aspect of visual abstraction, the ability to *produce* sketches spanning different levels of abstraction is no less important. Although there remains a gap between human and model sketch understanding, a CLIP-based vision model [70] was among the most performant and best aligned to human sketch understanding. Insofar as the major bottleneck to being able to generate more human-like sketches is achieving more human-like understanding of sketches and other images [24], a generative model leveraging CLIP's latent representation may be a promising approach. Consistent with this possibility, we found that CLIPasso generated sketches of concepts at each abstraction level were about as recognizable as the human-generated sketches of those same concepts at the same abstraction level (Fig. 6A; $adjusted\ R^2 = .64$). Moreover, we found that the more detailed CLIPasso sketches were especially human-like in that they evoked a similar set of meanings to their human-generated counterparts.

To measure the degree to which human and CLIPasso sketches converged with respect to the object labels that they elicited from human viewers, we computed the Jensen-Shannon distance (JSD) between the label distributions of human and CLIPasso sketches for each concept at each of the 4 levels of abstraction. In Fig. 6 B. we show the average label divergence at each level of abstraction. We found that humans and CLIPasso sketches were least divergent in terms of their perceived meaning when they were depicted in greater detail or were *less abstract* ($\beta = -0.69$, $t = -6.86$, $p < .001$). At lower levels of detail and visual fidelity, human and CLIPasso sketches elicited more diverging responses. Thus, while CLIPasso more closely approximates human sketch production behavior at greater levels of detail, there remains a large gap between how CLIPasso and human participants attempted to preserve sketch meaning under more severe production constraints. Taken together, while CLIPasso marks significant progress towards human-like sketch production its ability to produce highly abstract sketches in a human-like manner remains limited.

## 4 Conclusion

Recent advances in machine vision have enabled new opportunities to understand the computational mechanisms that underlie human-like visual abstraction—how humans create and interpret a wide variety of images (from detailed illustrations to schematic diagrams) to convey what they perceive and know about the world. Here we introduce SEVA, a new dataset and benchmark containing approximately 90K human and machine generated sketches spanning multiple levels of abstraction, to evaluate progress towards alignment between human and machine visual abstraction. Critically, our model evaluation protocol is focused not just on performance but on how well these models approximate *human-like*

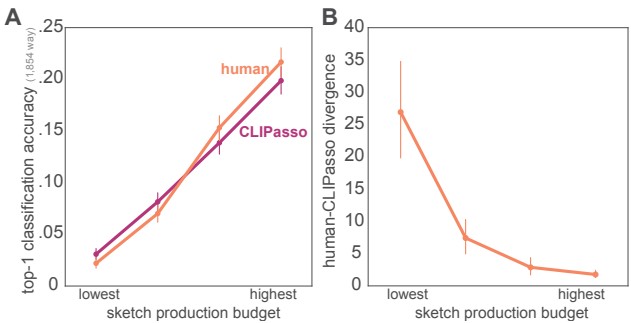

Figure 6: (A) Human top-1 recognition performance on CLIPasso and human sketches at different production budgets (draw duration for human sketches and number of strokes for CLIPasso sketches). (B) Divergence in human-responses to the same type of human and CLIPasso sketches as a function of sketch production budget.

sketch production and understanding. Initial benchmarking of a set of 17 vision models on SEVA following this protocol revealed that even the most performant and well-aligned models deviate from human behavior in systematic ways. We also find that current generative models of sketching are able to sketch in human-like ways, but only in limited settings.

We hope that SEVA will accelerate progress towards unified computational theories that explain how humans are capable of generating and understanding such a wide variety of abstract visual representations. Specifically, our dataset could be used to adjudicate between vision models that are high-performing on tasks involving natural images [64, 29] by characterizing their alignment with how humans understand images generated in a very different manner, including freehand sketches. In addition, SEVA is distinctive among existing human sketch datasets in that it contains detailed measurements of how humans make moment-to-moment decisions about where to place each stroke, when aiming to produce a sketch of a visible object, under different constraints. As such, we expect SEVA to be a key resource for advancing the state-of-the-art in sketch generation [69, 94, 91, 33, 72], and thus lead to computational models of generalized visual abstraction.

## 5 Limitations

We note several limitations of the current study that would be important to address in future work. First, our sample of participants was limited to English-speaking individuals based in the United States. As such, the current study cannot speak to potential differences in sketch-production behavior across geographical and cultural contexts. However, future work that recruits from a broader cross-section of individuals, including those located in the "Majority World," will be vital for understanding those potential sources of variation in human sketch production and comprehension. Second, we recruited participants via Prolific, a widely used crowdsourcing platform in human behavioral research, without regard to any previous artistic training they had received. As such, our study generally reflects sketch production behavior among individuals without substantial expertise in the visual arts. Third, following prior work [24, 23, 34, 105, 43, 59], the human sketches in our dataset were obtained using a web-based digital drawing interface, where most participants used a mouse or trackpad to produce their sketches (89.96%) and some participants used a touchscreen (6.94%) or a stylus (1.67%). Investigation of the impact of input device on sketch production would be a fruitful avenue for follow-up work. Fourth, because we did not obtain non-digital sketches (e.g., drawn on paper with a pen/pencil), our data cannot speak to differences between digital and non-digital sketches. Fifth, our study of machine sketch production included just one model, *CLIPasso*, limiting the conclusions that can be drawn about sketch generation algorithms in general. Future work that evaluates a broader suite of algorithms would thus be valuable.

## Acknowledgements

We thank members of the Cognitive Tools Lab at Stanford and the Stanford Neuro AI Lab for helpful feedback, discussions, and support. This work was supported by an NSF CAREER Award #2047191 to J.E.F.. J.E.F is additionally supported by an ONR Science of Autonomy award.

> All code and materials available at:
> https://github.com/cogtoolslab/visual_abstractions_benchmarking_public2023/

## Broader Impacts

Visual abstraction is as ubiquitous as it is central to our understanding of the visual world. Humans can recognize depictions of objects at varying levels of fidelity to their real-world counterparts. This ability to recognize and render concepts in abstract forms is crucial for visual knowledge transmission in the form of graphs, diagrams, and symbols. While computer vision has steadily made progress towards algorithms that can recognize objects in scenes, distinguish among different instances of an object, or even answer questions about those objects in natural language, it remains unclear if these systems have the capability to understand visual concepts at multiple levels of abstraction in a human-like manner. SEVA marks a step towards evaluating current vision algorithms on their sensitivity to depictions of a wide variety of common object concepts at varying levels of abstraction. Critically, we also provide results on *human* performance on sketch understanding for sketches of varying sparsity, setting a clear benchmark for future vision algorithms. Our initial study of state-of-the-art vision algorithms shows that even the most high-performing of models still fall short of human consistency baselines. With these vision algorithms increasingly being deployed in many aspects of everyday life, it is crucial to measure how human-like they are in this most natural of human abilities. We hope that SEVA will be used by the community to help close the gaps in current vision algorithms' alignment to human behavior that we have identified in this paper.

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
