# S Supplementary Materials

We confirm that the dataset will be available to the community under an MIT licence and that we bear all responsibility for the dataset and the associated codebase including in the case of violation of rights.

## S.1 Code and Materials

Code for reproducing the analyses reported in this submission as well as links to the dataset can be found here.

`https://github.com/cogtoolslab/visual_abstractions_benchmarking_public2023`

## S.2 Datasheet for Dataset

### S.2.1 Motivation

- **For what purpose was the dataset created?** To measure both humans and vision models in their ability to understand visual concepts at varying levels of abstraction.

- **Who created the dataset and on behalf of which entity?** The dataset was composed by the authors of this paper, and the data was generated by workers on Prolific and a generative sketching model, CLIPasso.

- **Who funded the creation of the dataset?** Funding for data collection was provided by NSF CAREER 2047191 to J.E.F.

### S.2.2 Composition

- **What do the instances that comprise the dataset represent?** Each instance consists of a digital sketch of one of 128 objects drawn under different time constraints. Some sketches were generated by a generative sketching model under different stroke constraints.

- **How many instances are there in total?** There are 89,797 individual human sketches. There are also 8,192 machine-generated sketches.

- **Does the dataset contain all possible instances or is it a sample of instances from a larger set?** Given the space of possible drawings at different abstraction levels is infinitely large, our dataset does not encompass every possible sketch instance. However, we provide many sketches of each concept at each abstraction level so as for the dataset to be representatively diverse.

- **What data does each instance consist of?** The sketch data consists of a *.png image of each sketch along with information about stroke trajectory, undo-history, the time-constraint or stroke-constraint under which it was produced, a label for the object it represents and which of 16 photographs of that object was shown to a participant when they were asked to draw it. Some sketches also have associated recognition data. This includes labels provided by human participants regarding what object they thought the sketch represented.

- **Is there a label or target associated with each instance?** Yes, each sketch is associated with a concept label and an abstraction level label.

- **Is any information missing from individual instances?** No, not that we are aware of. If we find any missing information we will update the dataset accordingly.

- **Are relationships between individual instances made explicit?** Yes, all relevant metadata to make connections between instances are provided.

- **Are there recommended data splits?** No.

- **Are there any errors, sources of noise, or redundancies in the dataset?** It is likely that some sketches might not be informative or might be noise. As we discover such instances, we will update our dataset accordingly.

- **Is the data self-contained, or does it link to or otherwise rely on external resources?** It is self-contained.

- **Does the dataset contain data that might be considered confidential?** No.

- **Does the dataset contain data that, if viewed directly, might be offensive, insulting, threatening, or might otherwise cause anxiety?** No.
- **Does the dataset relate to people?** Most of the sketches were generated by human participants, but otherwise no.

### S.2.3 Collection Process

- **How was the data associated with each instance acquired? What mechanisms or procedures were used to collect the data? How was it verified?** Sketches were produced by human workers on Prolific who were compensated for their time. The data were collected online using a JavaScript based sketching tool built in jsPsych. Recognition data were also collected on Prolific using another JavaScript-based tool also built in jsPsych. After accepting the task, participants provided informed consent in accordance with our university IRB and were provided instructions on how to do the task. Data were verified by the experimenters after data collection was completed

- **Who was involved in the data collection process and how were they compensated?** The data were collected by the authors of the paper and data were generated by Prolific workers. They were compensated at the rate of $15.50/hour according to the California minimum-wage.

- **Over what timeframe was the data collected?** All data were collected between January and May 2023 at different times.

- **Were any ethical review processes conducted?** All human data collection was approved by the University of California, San Diego IRB.

### S.2.4 Preprocessing/cleaning/labeling

- **Was any preprocessing/cleaning/labeling of the data done (e.g., discretization or bucketing, tokenization, part-of-speech tagging, SIFT feature extraction, removal of instances, processing of missing values)?** The raw data we present in our dataset were not put through any preprocessing pipelines. We did exclude data from participants who faced technical difficulties in our online task. A subset of our data were shown to a new cohort of Prolific workers for a recognition experiment and we provide data associated with that experiment in our dataset.

### S.2.5 Uses

- **Has the dataset been used for any tasks already?** Yes, the sketches produced in the human and machine sketch generation experiments were shown to participants to collect recognition data. In this paper, we also conduct several analyses on these data.

- **Is there a repository that links to any or all papers or systems that use the dataset?** No other papers use the dataset yet.

- **What (other) tasks could the dataset be used for?** The dataset could be used to finetune or adapt existing vision models and could also be used for human behavioral experiments that require participants to recognize abstract sketches.

- **Is there anything about the composition of the dataset or the way it was collected and preprocessed/cleaned/labeled that might impact future uses?** Not that we are aware of.

- **Are there tasks for which the dataset should not be used?** The authors are unaware of any use cases of this dataset that should be prohibited.

### S.2.6 Distribution

- **Will the dataset be distributed to third parties outside of the entity (e.g., company, institution, organization) on behalf of which the dataset was created?** Yes.

- **How will the dataset will be distributed?** It will be available using a combination of public Github repositories and links to raw human data on Amazon S3 instances.

- **When will the dataset be distributed?** During the time of the submission of this manuscript.

- **Will the dataset be distributed under a copyright or other intellectual property (IP) license, and/or under applicable terms of use (ToU)?** The dataset and associated code will be licensed under the MIT license. **Have any third parties imposed IP-based or other restrictions on the data associated with the instances?** No. **Do any export controls or other regulatory restrictions apply to the dataset or to indi- vidual instances?** No.

### S.2.7 Maintenance

- **Who is supporting/hosting/maintaining the dataset?** The dataset will be maintained by researchers at the Cognitive Tools Lab at Stanford University.

- **How can the owner/curator/manager of the dataset be contacted?** The corresponding author for this paper can be reached at their email. Links to the email will also be available on the public GitHub repository.

- **Is there an erratum?** Not presently.

- **Will the dataset be updated (e.g., to correct labeling errors, add new instances, delete instances)?** Yes, the dataset will be updated to correct for any errors.

- **Will older versions of the dataset continue to be supported/hosted/maintained?** Yes, as long as there aren't harmful or offensive sketches in the older versions.

- **If others want to extend/augment/build on/contribute to the dataset, is there a mechanism for them to do so?** We will provide code for our online sketching task and recognition task. If there is interest in building upon the dataset, the community will have the tools to do so. The authors welcome correspondence on this topic.

### S.3 Human Sketch Production Task

In each trial, participants were presented with a color photograph (500px x 500px) of an object paired with its concept label and asked to produce a drawing of the object concept in either 4 seconds, 8 seconds, 16 seconds, or 32 seconds. See Fig. 1 for an example task display.

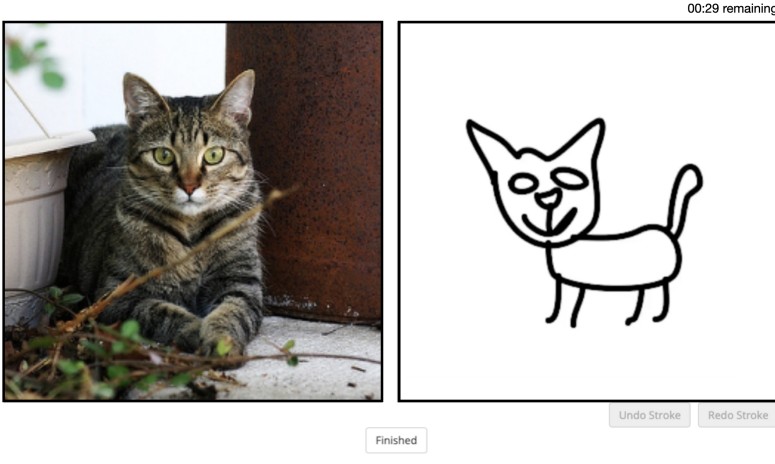

Figure 1: Web display for human sketch production task.

Prior to drawing, participants were told that the photograph cue was provided as a reminder of what the object looked like, but that they should make their drawings as recognizable as they could from the concept level, rather than of that specific example. They were encouraged to use as much time as they were permitted to produce as recognizable of a drawing as they could. The trial ended either when participants submitted their drawing or when time ran out.

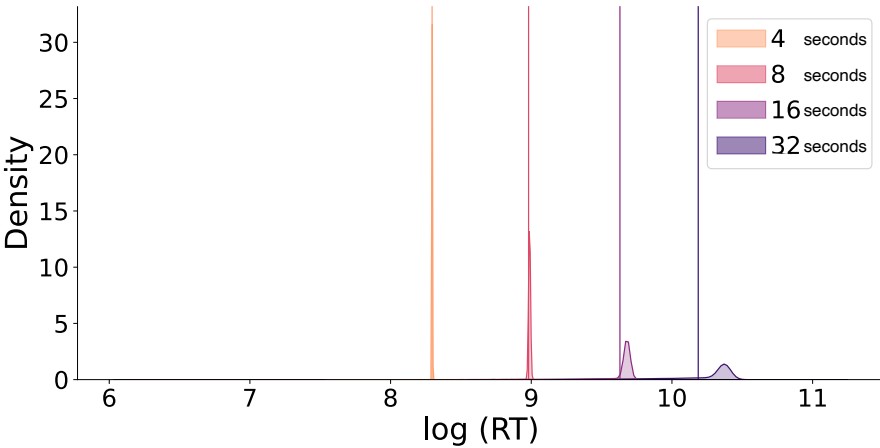

Figure 2: Distrubtion of Log response times (RTs) for each drawing constraint condition. Lines indicate mean log RTs.

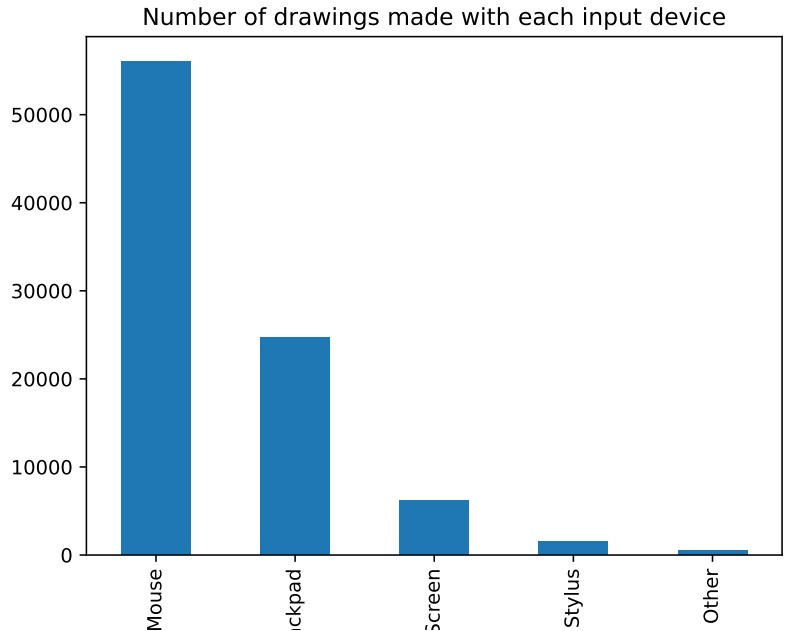

Figure 3: Number of sketches made using each unique input device.

## S.4   Human Sketch Recognition Task

On each trial of the recognition task, participants were presented with a single sketch on a canvas along with 3 blank text boxes.5. They were asked "What is this object?" and asked to provide at least 1 guess. Upon beginning to type their guess in any of the text boxes, a set of labels would appear in dropdown menu that contained matches from the entire set of 1,854 labels in the THINGS dataset to the current set of characters typed. Only responses that matched one of these 1,854 labels were accepted as valid.

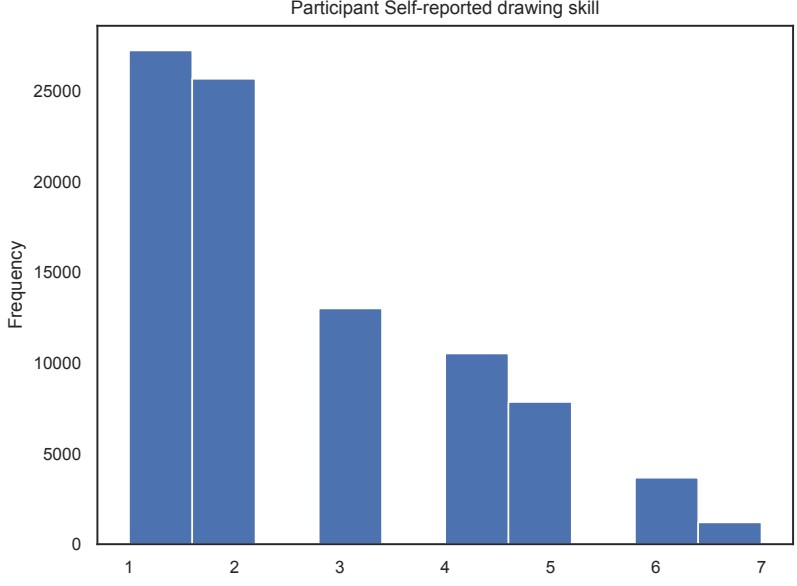

Figure 4: Distribution of participant reported drawing skill (on a scale from 1 to 7).

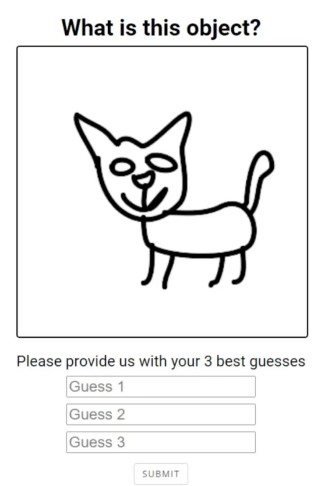

Figure 5: Web display for human sketch recognition task.

## S.5 Model Evaluation Protocol

### S.5.1 Extracting Class Probabilities from Vision Models

In this section, we provide a detailed description of the model evaluation protocol used in our study. For each pre-trained vision encoder in our model suite, we fit a regularized logistic regression model to predict the class label of each sketch based on the latent features of the model.

Following common practice in linear evaluation, we excluded the linear classification head from each pre-trained model and generated latent features using the rest of the encoder. In cases where the extracted features contained spatial dimensions (e.g., $height \times width \times channels$, or $tokens \times channels$), we computed the mean along the spatial dimensions, resulting in a 1-D vector feature for each sketch.

Then we used 5-fold stratified cross validation to assess the performance of the models. The human drawing dataset was divided into five splits, with the same number of $class \times production\_condition$ in each split. We fit a logistic regression model on each training split and used it to predict the labels of sketches in the corresponding test split. We used the logistic regression implementation from scikit-learn, with L2 regularization and L-BFGS optimizer.

Due to resource limitations, we performed a parameter search for the regularization strength on CLIP and VGG-19 models, over {0.01, 0.1, 1.0, 10.0, 100.0} Both models achieved the best performance with a regularization strength of 1.0. Therefore, we applied this parameter to the entire suite of models. We acknowledge that utilizing a fixed parameter may not yield the optimal performance for all models. In future versions of this paper, we plan to conduct individual parameter searches for each model to ensure the best possible performance across the board.

### S.5.2 Details On Semantic Neighbor Preference Score

In order to compute the semantic neighbor preference (SNP) score we first extracted semantic embeddings for each of the 1,854 THINGS concepts, which were previously computed by Hebart et al. (2023) based on human triplet similarity ratings. Next, for each concept we computed the rank of remaining 1,853 concepts in terms of their proximity to to the target concept. Determining the SNP score for sketches of a given concept at a given level of abstraction involved obtaining the ranks of all the top-1 guesses of all the sketches of that type and excluding sketches where the guess matched the ground-truth. This was because the SNP score measures how semantically related *off-target* labels are to the true label. We next computed the cumulative proportion of labels provided for that kind of sketch as a function of rank. Thus, if most guesses were semantic neighbors of the true label then the cumulative proportion would increase and reach a value of 1 quickly. In contrast, if many of the guesses were semantically distant from the true label, their ranks would be much lower and thus the cumulative proportion would also increase slowly. Fig. 6 shows how this cumulative proportion of labels varies as a function of rank (normalized from 1-1853 to 0-1) averaged over all sketch types. The SNP score is the area under the curve (AUC) in this plot such that an AUC value closer to 1 indicates that guesses are generally in the same semantic neighbourhood as the ground-truth label, an AUC value of 0.5 indicates that labels are random and an AUC value close to 0 indicates that guesses are systematically far away from the ground truth.

### S.5.3 Details On Fitting Linear Regression Models

Standard ordinary least squares regression models were fit using the statsmodels package in Python.

Mixed-effects linear regression models reported in this paper were fit in R using the lme4 package version 1.1.30. We generally made use of the bobyqa optimizer when fitting mixed-effects models.

**Humans produce sparser sketches under stronger time constraints**. We fit models predicting the number of strokes from the drawing duration (42, 82, 16s, or 32s). We included random intercepts and slopes for the effect of drawing time for each concept.

**Sparser sketches are more semantically ambiguous for models and humans** We fit separate sets of models for predicting human and vision model performance. For human performance we fit 3 regression models predicting top-1 classification accuracy, response entropy, and semantic neighbor preference with drawing duration as the only predictor in each. We included random intercepts and slopes for the effect of drawing time for each concept.

For vision model performance we also fit 3 separate models predicting top-1 classification accuracy, response entropy, and semantic neighbor preference with drawing duration and model type as predictors. Model-type was a dummy-coded variable coding for each of the 17 vision models with CLIP as the reference level.

**Different models display distinct patterns of sketch recognition behavior**

To test whether top-1 classification accuracy, response entropy, and semantic neighbor preference varied across each of the 17 vision models, we also fit reduced versions of the models described in the earlier section *without* a predictor for model-type. We next performed a $\chi^2$ model comparison between the full and reduced models to test whether the amount of variance explained by the regres-

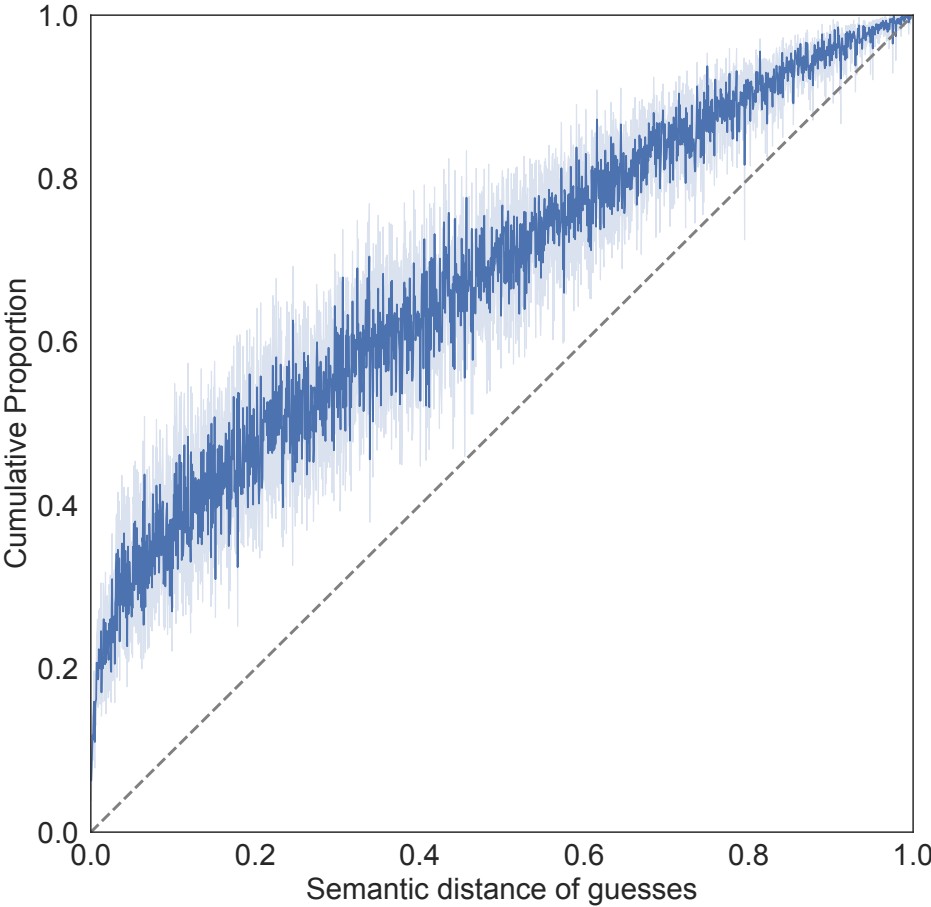

Figure 6: Cumulative proportion of label responses in the human recognition experiment as a function of semantic distance of the labels from the ground-truth label averaged over different concepts and abstraction levels.

sion models that did include the model-type predictor was significantly different from the regression models that did not include the predictor.

**A large gap remains between human and model sketch understanding**

We estimated human-baselines by splitting the human data in half multiple times and fitting linear regression models predicting top-1 classification accuracy, response entropy, and semantic neighbor preference values of one half of the data from the other half with an interaction term for draw duration to account for variance within each 'abstraction level'. We report the mean of the adjusted model $R^2$ values across multiple splits of the data as the baseline values.

When computing human-model alignment we also fit linear regression models predicting vision-model top-1 classification accuracy, response entropy, and semantic neighbor preference values from the human values for the corresponding sketches. Here too, we included an interaction term for draw duration in each regression model. We report adjusted $R^2$ as a measure of human-model alignment. To compute confidence intervals we sampled the dataset of all sketches multiple times with replacement, generating bootstrapped samples of the same size as the original dataset. We bootstrapped samples within each concept and draw duration level to ensure that each sample was balanced in terms of the number of sketches of each concept at each level of abstraction.

**A CLIP-based sketch generation algorithm emulates human sketches under some conditions**

In order to compare human top-1 classification of human-made and `CLIPasso`-generated sketches, we fit a linear regression model predicting human top-1 classification accuracy from `CLIPasso`

classification accuracy of the same type of sketches. We included an interaction term for abstraction level, similar to earlier analyses. We report the adjusted $R^2$ value of the model as a measure of how comparably recognizable human and `CLIPasso` sketches were.

*Human-CLIPasso response divergence.* To measure the degree to which human label responses diverged between human-made and `CLIPasso`-generated sketches of the same concept at the same level of abstraction, we adopted the following procedure. For each type of machine sketch (e.g., lion made with 4 strokes), we computed the Jensen-Shannon distance (JSD) between that type of machine sketch and *all* types of human sketches (4s lions, 8s lions, ... 16s tanks, 32s tanks). Next, we ranked all the human sketches in terms of their proximity to the machine sketch and recorded the minimum rank for human sketches belonging to the same concept. For example, if the machine sketch was a 4-stroke lion and the ranks of the 4s, 8s, 16s, and 32s human lion sketches were 7,13,39, and 71 respectively then we would record **7**, the rank corresponding to the 4s human sketch. This number is our reported measure of human-CLIPasso response divergence. In essence, our measure answers the question - Do *any* human lion sketches elicit similar labels as a 4-stroke `CLIPasso` lion sketch?