# OpenReview forum: "SEVA: Leveraging sketches to evaluate alignment between human and machine visual abstraction"
_NeurIPS.cc/2023/Track/Datasets_and_Benchmarks — NeurIPS 2023 Datasets and Benchmarks Poster_

### Official Review · Reviewer_81P5 · 2023-07-05
**an interesting dataset containing large-scale human sketches and assessment of human-model alignment**

**Rating:** 7
**Confidence:** 4
**Correctness:** All the claims are correct to me.
**Clarity:** The writing is good and easy to follow.

**Strengths:**

1. The paper is generally well-written.
2. The problem of measuring sketch alignment and recognition between ai and humans is an important and interesting problem.
3. Multiple aspects of alignments are covered and assessed rigorously. This includes introduction to multiple evaluation metrics, and benchmarking 17 models.
4. The dataset design is well-structured and well-organized (see some suggestions and questions below).

**Additional Feedback:**

1. The quality of data collected from online platforms, such as prolific and mturk, is not always good. Are there any quality control measures taken by the paper when collecting human classification responses and human sketches? How does the paper filter out bad responses or bad drawings?

2. In Fig5, I noticed that CLIP is generally better than transformers, than convnets, any intuitions behind this? Does the trend here align with recogniton accuracies on ImageNet as well?

3. In Fig5, A-C can also be provided for the sketches generated by CLIPasso. This is for easy comparsion with the human generated sketches.

4. In Fig7B: how is the divergence calculated? Does it correspond to line 351 - 359?

5. When humans perform sketching tasks, the dynamics of the strokes vary, such as the velocity and the momentum. Can the paper comment on this? How would dynamics play a role in model recognition (if applicable) and how would this differ from CLIPasso (if applicable)?

6. Could the paper also consider benchmarking recurrent models (such as lstm, GRU) for recognition tasks as well?

**Documentation:**

The paper promises to release the code and data.

**Ethics:**

No ethical concerns, as far as I can see.

**Limitations:**

See the limitations and suggestions below.

**Opportunities For Improvement:**

See the limitations and suggestions below.

**Relation To Prior Work:**

The paper surveys a sufficient number of datasets and baseline methods.

**Summary And Contributions:**

The paper proposes a large-scale dataset containing human drawings with different sparsity levels. Some interesting insights are obtained by comparing different recognition models and by comparing the sketches generated by humans and CLIPAsso.

---

> ### Author Response · Authors · 2023-08-21
> **Response to Reviewer 81P5 (part 1)**
>
> We thank the reviewer for taking the time to read our paper and for the helpful comments.
> Below we address the concerns raised:
>
> > The quality of data collected from online platforms, such as prolific and mturk, is not always good. Are there any quality control measures taken by the paper when collecting human classification responses and human sketches? How does the paper filter out bad responses or bad drawings?
>
> We specifically chose to use Prolific, which has been shown to have much higher data quality relative to mTurk [1]. Secondly, we compensated participants at a competitively high rate @ $15.5/hr, thereby placing participants in a circumstance where it would be possible to provide higher quality data (vs. needing to rush through a large volume of low-paying studies). In comparable sketch production experiments at similar scales, we have found that less than 4% of drawings produced were judged to be invalid by a set of 3 validators. Thus, we expect our sketch dataset to be of high quality. We also included a ‘catch-trial’ for the recognition experiment to provide a basic measure of task engagement, where the participant was expected to provide the label ‘cat’ to a fairly unambiguous sketch of a cat. We excluded participants who failed to correctly identify this sketch.
>
> > In Fig5, I noticed that CLIP is generally better than transformers, than convnets, any intuitions behind this? Does the trend here align with recognition accuracies on ImageNet as well?
>
> Thank you for making this observation, which we also find quite intriguing! While CLIP is higher than other Transformers and ConvNets for some metrics (top-1 classification accuracy alignment), this is not always the case (e.g. see semantic neighbor preference alignment). Additionally recent work has begun to show, albeit not in the domain of sketch understanding, that as vision models become more performant on ImageNet metrics, they become less aligned with human behavior [2]. Thus, in the main paper we have opted to avoid drawing conclusions about the impact of model architecture without conducting additional experiments that control for other important factors, such as the training dataset size/distribution.
>
> > In Fig5, A-C can also be provided for the sketches generated by CLIPasso. This is for easy comparsion with the human generated sketches.
>
> Thank you for suggesting this interesting idea for extending our analysis approach! And we believe that there is a version of this evaluation that would be valuable to pursue in future work with more thorough & controlled experiments (i.e., including a broader variety of sketch generation algorithms, under conditions that control for training data leakage between sketch generation and recognition). **We now note the value of such studies explicitly in our Limitations section**.
>
>
> References:
>
> [1]  Douglas, B. D., Ewell, P. J., & Brauer, M. (2023). Data quality in online human-subjects research: Comparisons between MTurk, Prolific, CloudResearch, Qualtrics, and SONA. Plos one, 18(3), e0279720.
>
> [2] Fel, T., Rodriguez Rodriguez, I. F., Linsley, D., & Serre, T. (2022). Harmonizing the object recognition strategies of deep neural networks with humans. Advances in Neural Information Processing Systems, 35, 9432-9446.

---

> > ### Author Response · Authors · 2023-08-21
> > **Response to Reviewer 81P5 (part 2)**
> >
> > > In Fig7B: how is the divergence calculated? Does it correspond to line 351 - 359?
> >
> > We apologize for the confusion. For each type of machine sketch (e.g., lion made with 4 strokes), we computed the Jensen-Shannon distance (JSD) between that type of machine sketch and all types of human sketches (4s lions, 8s lions, ... 16s tanks, 32s tanks). Next, we ranked all the human sketches in terms of their proximity to the machine sketch and recorded the minimum rank for human sketches belonging to the same concept. For example, if the machine sketch was a 4-stroke lion and the ranks of the 4s, 8s, 16s, and 32s human lion sketches were 7,13,39, and 71 respectively then we would record 7, the rank corresponding to the 4s human sketch. This number is our reported measure of human-CLIPasso response divergence. **We address how divergence is computed in our supplemental materials (L217) due to space constraints**.
> >
> >
> > >When humans perform sketching tasks, the dynamics of the strokes vary, such as the velocity and the momentum. Can the paper comment on this? How would dynamics play a role in model recognition (if applicable) and how would this differ from CLIPasso (if applicable)?
> >
> > A critical facet of SEVA is that we do record and preserve the stroke trajectories for each sketch as metadata. While here we only evaluate models’ abilities to recognize complete sketches, one could imagine ablating strokes and testing models on their performance and alignment to human behavior. We might expect that more detailed sketches would be more robust to ablations than sparser sketches. While CLIPasso does not generate its strokes in a sequential manner, our hope is that with the help of datasets like SEVA, future models of sketch production will not only produce abstract sketches that look like human sketches but also produce them in the same way as humans do.
> >
> > >Could the paper also consider benchmarking recurrent models (such as lstm, GRU) for recognition tasks as well?
> >
> > This is a great suggestion! While we believe that sequential models are a great candidate for benchmarking understanding of stroke-by-stroke graphical productions such as sketches, comparing these models to our current model suite in a fair way would require adapting our evaluation methods in a way that is not immediately obvious and non-trivial to conduct within the review period. That said, we hope that by publicly releasing our dataset it will be easier for the research community to collectively make progress on performing a more comprehensive set of model evaluations!

---

> > > ### Comment · Reviewer_81P5 · 2023-08-21
> > > **Thank you for the responses from the authors**
> > >
> > > I have read the responses from the authors as well as comments from other reviewers. The responses of the authors have clarified my initial doubts. My rating stands the same - Accept

---

### Official Review · Reviewer_K3MZ · 2023-07-21
**An interesting paper with room for improvement**

**Rating:** 6
**Confidence:** 5
**Correctness:** The paper seemed to follow a logical …
**Clarity:** Writing seemed clear to me.

**Strengths:**

I appreciate the authors efforts for this paper and found multiple strengths as follow:

1. **[Data gathering]** The data gathering process was clearly explained.
2. **[Motivation & insight]** The motivation behind understanding how humans perceive different levels of abstraction is a valuable research direction. This research direction might help to better understand human abstraction and perception. It might also help to develop algorithms that might encourage machines to have similar abstraction levels as humans or human-like deep learning models.
3. **[Providing multiple labels for ambiguous sketches]** It was a smart direction to provide multiple labels for ambiguous sketches because it can help to design better classifiers in future for sketch recognition, which can model these ambiguities.
4. **[Various baselines]** Authors incorporated a considerable number of baselines (17 models), which is great in understanding each model's capabilities to deal with various abstraction levels and ambiguities.
5. **[Valuable data]** It is hard to gather human drawn sketches in a large scale because it requires lots of effort and participants. Therefore, I appreciate the effort used to gather this data.


**Additional Feedback:**

I stay positive for this paper, and I hope authors further improve the current work.

**Documentation:**

Documentation was provided using "Datasheet for Dataset" framework.

**Ethics:**

I am not an expert in ethics but it seemed that authors considered ethics for conducting their user studies and data gatherings.


**Limitations:**

**[Drawing tool might impact the human sketch abstraction]** Participants may use any device for drawing sketches with different abstraction levels. Sometimes it might be hard to draw a sketch if a participant only uses a computer mouse. Then, the drawn sketch is not representing human visual abstraction level but it might also get influenced by the drawing device. For example, if all participants were provided with an electric pen and a tablet, the drawn sketches might look completely different than ones which are only generated by a computer mouse.

**Opportunities For Improvement:**

I think the following might help to better show the value of the proposed dataset:

**[Providing a more tangible use of the proposed dataset]** While I appreciate the dataset value for analyzing human visual abstraction, it will be great if authors can provide some clear examples or uses of the dataset. As mentioned in the paper, if the ultimate goal is to generate machine learning models that have human-like visual abstraction, how should we use this data to reach this goal? For example, is there any specific way that we use this data to increase the machine performance in understanding more abstracted sketches?


**[Clarifying the need for human-like models]** To clarify the motivation behind this paper, why do we need to develop models that are human-like? Sometimes machines can perform better than humans in some tasks, and I think there is no scientific proof that humans have the optimal visual understanding or abstraction. This paper might help to understand why humans sometimes perform better in generalization tasks on unseen data, but it does not provide any systematic way or insight to solve this problem. Therefore, I think clarifying why we need to develop human-like models (e.g., for better generalization on unseen data) with references from the literature might better clarify final goal than a more general statement of developing a human-like models is needed.

**Relation To Prior Work:**

I appreciate the author's effort to provide the prior work. There are some recent sketch related papers in the computer graphic society (e.g., Transactions on Graphics (TOG), SIGGRAPH, SIGGRAPH Asia) that are missing here and can be added for the completeness.

**Summary And Contributions:**

This paper introduces a dataset containing 90K human sketches and synthetic machine generated sketches with different levels of abstraction (sketch sparsity).

To generate different abstraction levels, the paper asked participants to draw sketches under time constraints (e.g., 4  or 8 seconds), which resulted in sketches with different sparsities. A machine learning approach (CLIPasso) was also used to generate synthetic sketches with different sparsity based on the number of strokes.

Human sketch understanding was examined by showing these various sparse sketches to people and asking them to define each sketch class. Machine sketch understanding was also examined by evaluating the performance of pre-trained models for automatically detecting each sketch class.

Finally, the differences and similarities between human perception and deep learning models' inference are concluded (Section 3). While some conclusions seemed really obvious (e.g., Sparser sketches are more semantically ambiguous for both models and humans), the paper scientifically proved these statements, which are important to better understand human perception of different abstraction levels as well as human-like sketch understanding and generation.

---

> ### Author Response · Authors · 2023-08-21
> **Response to Reviewer K3MZ (part 1)**
>
> We thank the reviewer for taking the time to read our paper and for sharing such thoughtful feedback. Below we provide our responses in a point-by-point manner:
>
> > [Providing a more tangible use of the proposed dataset] While I appreciate the dataset value for analyzing human visual abstraction, it will be great if authors can provide some clear examples or uses of the dataset. As mentioned in the paper, if the ultimate goal is to generate machine learning models that have human-like visual abstraction, how should we use this data to reach this goal? For example, is there any specific way that we use this data to increase the machine performance in understanding more abstracted sketches?
>
> Thank you for the opportunity to clarify the potential use cases for the dataset beyond the results we present in this paper. In our revision, we now include a new paragraph in the Conclusion section that addresses this issue. In short, we propose that SEVA can be used to test new vision models on how well their performance aligns with human performance using the broader suite of evaluation metrics used in the paper. In particular, we propose that a crucial adjudicating factor between models should be how aligned their behavior is with human image understanding behavior (vs.  task performance alone), and that sketches are a fruitful domain for differentiating vision models, owing to large differences between sketches and natural images. Second, we envision SEVA also being a resource for developing new models of sketch generation, given the availability of stroke-level information and human sketches produced at different levels of abstraction. **We have elaborated this point in our revised Conclusions section L386-397**.
>
> > [Drawing tool might impact the human sketch abstraction] Participants may use any device for drawing sketches with different abstraction levels. Sometimes it might be hard to draw a sketch if a participant only uses a computer mouse. Then, the drawn sketch is not representing human visual abstraction level but it might also get influenced by the drawing device. For example, if all participants were provided with an electric pen and a tablet, the drawn sketches might look completely different than ones which are only generated by a computer mouse.
>
> Thank you for raising this point, which was also expressed by another reviewer (“p23D”).
>
> Our overarching goal was to develop a benchmark of human sketch production and understanding that is: (1) large-scale; (2) broadly inclusive; (3) compatible with previous sketch datasets; and (4) provide a strong basis for follow-on work. As such, we recruited participants online (rather than in-person, which would limit sampling to participants physically located near the researchers’ institution). In addition, we did not require them to use any particular input device, as imposing such a requirement would impede the ability for a large fraction of potential participants to contribute (more than 89% of participants used a mouse or trackpad). Further, we opted to focus on sketches produced using a digital interface (and often using a mouse or trackpad), in line with much previous work developing sketch benchmark datasets (see references cited at the beginning of Section 2.1). Finally, we approached data collection with an eye to valuable follow-on work, including work that directly investigates the impact of input device on sketching behavior. Given the prevalence of digital devices in the 21st century, we believe it is valuable to thoroughly characterize how humans generate digital images. However, we agree with the reviewer that it is also important to understand how drawing behavior might differ systematically when using different input devices or when producing a sketch using a digital interface vs. pen/paper. Such comparisons would be a fruitful avenue for future work.
>
> **In our revision, we now explicitly note the proportion of participants who used each form of input device in our study, as well as limitations of our approach, and the value of future work that investigates the association between input device and sketching behavior (see Limitations section)**.

---

> > ### Author Response · Authors · 2023-08-21
> > **Response to Reviewer K3MZ (part 2)**
> >
> > > [Clarifying the need for human-like models] To clarify the motivation behind this paper, why do we need to develop models that are human-like? Sometimes machines can perform better than humans in some tasks, and I think there is no scientific proof that humans have the optimal visual understanding or abstraction. This paper might help to understand why humans sometimes perform better in generalization tasks on unseen data, but it does not provide any systematic way or insight to solve this problem. Therefore, I think clarifying why we need to develop human-like models (e.g., for better generalization on unseen data) with references from the literature might better clarify final goal than a more general statement of developing a human-like models is needed.
> >
> > Thank you for prompting us to clarify the scientific motivation behind this paper and the potential implications for machine learning. We see the development of scientific theories that explain the nature of human intelligence as a fundamental research target in advancing a broader science of intelligence. It is true that for many tasks (e.g., bar code reading), humans fall short of other species of intelligent systems (natural or artificial). Nevertheless, as a complete package, given humans’ remarkable capacity for continual learning and generalization, social organization, among other abilities, they are clearly among the most worthy targets of investigation. Making the full argument for the importance of developing computational theories of human intelligence goes beyond the scope of these comments and our single paper, which must of course be scoped to address a concrete problem. The concrete problem we have chosen is visual abstraction, a suite of abilities demonstrating flexible production and comprehension of visual images, one that does not seem to be shared with other known intelligent species, and one that lies at the foundation of key human inventions, including writing, mathematics, and art. For such a suite of abilities, we propose that there is no simple way to assess “optimality” such that a system could “outperform” humans. Rather, our contention is that building algorithms that emulate human-like visual abstraction is a worthy goal because such algorithms would constitute scientific theories of an important phenomenon.
> >
> > Nevertheless, we agree with the reviewer that it is also important to consider the implications of this work for machine learning. **As such, in our revision we have now included new text (end of Section 1.1: Human Visual Abstraction as Key Target for AI) that makes the case for incorporating principles from human vision, based on specific prior studies that have found that such approaches have led to more robust vision algorithms**.
> >
> > We also wish to clarify that visual abstraction, as operationalized in our paper, goes beyond generalizing to new classes of images. Rather, visual abstraction also means being able to exhibit different degrees of uncertainty when faced with a detailed vs. abstract sketch. And the extent to which the uncertainty expressed by any AI system aligns with human uncertainty is taken to be a key measure of how capable that system is of visual abstraction. **We emphasize this in L92 noting that accuracy metrics alone do not capture visual understanding across abstraction levels**. In line with work promoting the importance of incorporating uncertainty into AI systems to improve robustness and explainability [1][2][3], here we extend this idea to say that the uncertainty expressed should also be human-like.
> >
> > References:
> >
> > [1] Peterson, J. C., Battleday, R. M., Griffiths, T. L., & Russakovsky, O. (2019). Human uncertainty makes classification more robust. In Proceedings of the IEEE/CVF International Conference on Computer Vision (pp. 9617-9626).
> >
> > [2] Collins, K. M., Bhatt, U., & Weller, A. (2022, October). Eliciting and learning with soft labels from every annotator. In Proceedings of the AAAI Conference on Human Computation and Crowdsourcing (Vol. 10, No. 1, pp. 40-52).
> >
> > [3] Bhatt, U., Antorán, J., Zhang, Y., Liao, Q. V., Sattigeri, P., Fogliato, R., ... & Xiang, A. (2021, July). Uncertainty as a form of transparency: Measuring, communicating, and using uncertainty. In Proceedings of the 2021 AAAI/ACM Conference on AI, Ethics, and Society (pp. 401-413).
> >
> >
> >
> > > I appreciate the author's effort to provide the prior work. There are some recent sketch related papers in the computer graphic society (e.g., Transactions on Graphics (TOG), SIGGRAPH, SIGGRAPH Asia) that are missing here and can be added for the completeness.
> >
> > Thank you for pointing out gaps in our review of related work. **In our revision, we have added several new references (L65-69) covering sketch categorization, segmentation, sketch-based retrieval, and generation. We have also included a new table (Table 1 on page 3) that lists earlier sketch datasets and their properties**.

---

> > ### Comment · Reviewer_K3MZ · 2023-08-28
> >
> > I appreciate the authors' effort for revising the paper. I read your comments multiple times, and I am thankful for the provided clarifications.
> >
> > While I appreciate your effort to improve the paper, I still think that the drawing tool might completely impact the abstraction level. For example, when someone uses a mouse, it might cause more abstracted sketch than using a pen. Therefore, it is better to report conclusions of the paper on the data which is only gathered using one tool.
> >
> > Because this track looks for the wider impact of a dataset, I still do not know how one can use this data to train a model that can have a better generalization like humans. I mean I cannot clearly see a tangible future work from this dataset that was not possible with previous ones. In this paper, the data is mostly used for analysis, resulting in some obvious conclusions (for example, sparser sketches are more semantically ambiguous). Therefore, I am not sure how one can use this data for further development of algorithms that can improve AI models for the wider impact of this dataset.
> >
> > I appreciate that authors added citations for supporting their claim as "Human Visual Abstraction as Key Target for AI" because now it is technically more sound than a general statement.
> >
> > As a minor comment, there are still various missing references from the Computer Graphics Community that can be easily found by searching "sketch" in the title of the Transaction on Graphics (TOG) or SIGGRAPH or SIGGRAPH Asia papers.
> >
> > In summary, I will stay in the positive side for this paper like the previous time but I still believe that there is room for improvement, which can make this paper to be even more impactful.

---

### Official Review · Reviewer_p23D · 2023-07-21

**Rating:** 7
**Confidence:** 4
**Clarity:** The paper is well written and easy to…

**Strengths:**

+ This paper contributes a large-scale human-generated sketch dataset with a large number of participants.

+ The key findings from the benchmark study are very interesting, such as the "none of the models came close to approximating human response patterns", "a gap between how sketch generation algorithm CLIPasso [62] and human participants attempted to preserve sketch meaning", etc. These findings could be useful and inspire corresponding research.

+ The findings concluded in the Results section are interesting but intuitively make sense. These might be useful for following-up sketch-related research.

+ The paper is well-written and clearly motivated. The reviewer enjoyed reading it, and believes it will inspire the audience of the conference.

**Additional Feedback:**

It would be helpful if the authors could address the concerns mentioned above in the "Opportunities For Improvement" section.
It would also be helpful if more examples of different instances under the same category could be shown, e.g. in Fig. 3.

**Correctness:**

The dataset was generally constructed in a sound way if the above-mentioned concerns could be well addressed. In the benchmark evaluation, the methods and experimental designs are appropriate and performed correctly. The authors also find a good way (using those three metrics) to evaluate the performance.

**Documentation:**

There is sufficient detail on data collection and organization in general. The availability and maintenance details were also provided. The intended uses were also discussed in the supplementary material.
For the benchmark, there seems to be sufficient detail to support reproducibility, considering the compared methods are all published methods with code available.

**Ethics:**

There are no clear ethical concerns with the submission that warrant further discussion or review.

**Limitations:**

The authors did not adequately address the limitations and potential negative societal impact of their work, though they mentioned broader impacts in a more positive way.
Possible limitations were mentioned a bit in the above sections, e.g. the way the human-generated sketches were collected.
The potential societal impact could be the bias built in the data collection part and the corresponding influence on the findings and algorithm design.

**Opportunities For Improvement:**

- It is unclear why "each participant was randomly assigned to one of four conditions" (L152) instead of drawing with all the four time conditions, to consistently show the difference among the conditions. If randomly assign to only one of those four time conditions, the bias within each participant may lead to some issues in analysing the four conditions.

- From the description of the sketch production, the participants drew the sketches "using whatever input device they had". Would this cause any inconsistency or even unfair analysis? For example, usually sketches drawn by a stylus are more natural than those drawn by a mouse.

- From the description of the sketch drawing, the participants may continue to the next trial before the pre-defined time was used up, e.g. one may only use 10 seconds for a 32 seconds trial, and many other variations. As one of the key points this study looks into is the difference between different conditions/time, would this cause any issue or inaccuracy? A distribution of the actual time used for each participant under each pre-assigned condition would possibly show a better understanding of such an effect.

- For the human sketch understanding task (Sec. 2.2), 1,709 participants were recruited for this study. But it is unclear if there were any overlap with the previous 5,563 participants in the sketch production (Sec. 2.1). If there are same participants between these two groups, it may lead to some data/knowledge/prior leakage issues.
Similarly, it is unclear if the later 1,481 participants in the "human understanding machine sketches" study have any overlap with the above groups. More details should have been clarified.

- Missing reference for the statement "One study found that 74 current vision models trained on photorealistic image..." (L74).

**Relation To Prior Work:**

The differences between this work and previous contributions were clearly discussed, but it would be better if a summarisation table could be included to show the key differences and relations between this work and existing works from different perspectives.

**Summary And Contributions:**

This paper presented a benchmark study for alignment evaluation between human and machine visual abstraction, in terms of sketch representation. Specifically, a new large-scale sketch dataset was introduced, containing 90K human-generated sketches of 128 object concepts. A thorough analytical study covering 17 vision models and the human understanding shows several interesting observations and findings. The main contributions of this work are the newly introduced large-scale sketch dataset and the corresponding analysis between human and machine visual abstraction.

---

> ### Author Response · Authors · 2023-08-21
> **Response to Reviewer p23D (part 1)**
>
> We thank the reviewer for taking the time to read our paper and for their several insightful comments. Below we address each comment in a point-by-point manner:
>
> > It is unclear why "each participant was randomly assigned to one of four conditions" (L152) instead of drawing with all the four time conditions, to consistently show the difference among the conditions. If randomly assign to only one of those four time conditions, the bias within each participant may lead to some issues in analysing the four conditions.
>
> Thank you for prompting us to clarify the rationale behind our use of a “between-subjects design” in our human experiment. The key idea is randomization — that is, random assignment of individual participants to drawing-time condition ensures that any reliable individual differences in sketch production behavior (i.e., “bias within each participant”) will not, in expectation, bias our estimates of differences between conditions, especially in a sample as large as the one we obtained.
> The salient alternative to a “between-subjects design” is a “within-subjects design,” wherein every individual participant produces sketches in all drawing-time conditions. The main issue with this approach is the high likelihood of “order effects,” a pervasive form of non-independence when obtaining repeated measurements of human behavior. For example, the condition participants were first exposed to might influence their drawing behavior on subsequent trials belonging to other conditions. Such directional influences would introduce bias in our estimates of differences between conditions. A second issue concerns the impact of “switch costs,” a well documented feature of human sequential behavior, where humans might adopt a sketch production strategy that is appropriate for one drawing-time condition, and the time taken to fully switch to a strategy appropriate to a new drawing-time condition exceeds the brief interval between trials.
>
> Given these considerations, we reasoned that a between-subjects design would be more appropriate to address the core objectives of our benchmarking study. **In our revision, we clarify the rationale for our choice of experimental design (lines 156-158)**.
>
> > From the description of the sketch production, the participants drew the sketches "using whatever input device they had". Would this cause any inconsistency or even unfair analysis? For example, usually sketches drawn by a stylus are more natural than those drawn by a mouse.
>
> Thank you for prompting us to clarify the rationale for allowing participants to use any input device they had, and the implications of this design choice. First, our goal was to develop a benchmark of human sketch production and understanding that is: (1) large-scale; (2) broadly inclusive; (3) compatible with previous sketch datasets; and (4) provide a strong basis for follow-on work. As such, we recruited participants online (rather than in-person, which would limit sampling to participants physically located near the researchers’ institution). In addition, we did not require them to use any particular input device, as imposing such a requirement would impede the ability for a large fraction of potential participants to contribute (more than 89% of participants used a mouse or trackpad). Finally, we approached data collection with an eye to valuable follow-on work, including work that directly investigates the impact of input device on sketching behavior. Given the prevalence of digital devices in the 21st century, we believe it is valuable to thoroughly characterize how humans behave when using digital media. However, we agree with the reviewer that it is important to understand how drawing behavior might differ systematically when using different input devices or when producing a sketch using a digital interface vs. pen/paper.
>
> **In our revision, we now explicitly note the proportion of participants who used each form of input device in our study, as well as limitations of our approach, and the value of future work that investigates the association between input device and sketching behavior (see Limitations section)**.

---

> > ### Author Response · Authors · 2023-08-21
> > **Response to Reviewer p32D (part 2)**
> >
> > > From the description of the sketch drawing, the participants may continue to the next trial before the pre-defined time was used up, e.g. one may only use 10 seconds for a 32 seconds trial, and many other variations. As one of the key points this study looks into is the difference between different conditions/time, would this cause any issue or inaccuracy? A distribution of the actual time used for each participant under each pre-assigned condition would possibly show a better understanding of such an effect.
> > For the human sketch understanding task (Sec. 2.2),
> >
> > Thank you for raising this question. The critical test of whether our manipulation of drawing time succeeded concerns the observed impact on the amount of detail in each sketch. We note that **Figure 3** strongly suggests that our manipulation succeeded, with drawings made in the 32 second condition containing the greatest number of strokes (proxy for amount of detail) and sketches made in the 8 second condition having the fewest strokes. **Our statistical analyses provide rigorous evidence reinforcing the impression from Figure 3 (see L269)** .
> >
> > Nevertheless, to enhance transparency, we agree with the reviewer that it would be helpful to also show the distribution of actual drawing times in each condition. **As such, we now include an additional figure in the supplementary materials (Supp. Fig. 2) that plots the distributions of actual response times for each of the 4 conditions**. Here too we can see a strong association between drawing-time condition and actual drawing time (the distributions are non-overlapping).
> >
> > > 1,709 participants were recruited for this study. But it is unclear if there were any overlap with the previous 5,563 participants in the sketch production (Sec. 2.1).
> > If there are same participants between these two groups, it may lead to some data/knowledge/prior leakage issues. Similarly, it is unclear if the later 1,481 participants in the "human understanding machine sketches" study have any overlap with the above groups. More details should have been clarified.
> > Missing reference for the statement "One study found that 74 current vision models trained on photorealistic image..." (L74).
> >
> > Thank you for this very helpful comment! Between the 5,563 participants recruited for the sketch production study and the 1,709 participants recruited for the human sketch recognition study, there were 579 participants who did both studies. However, no sketch produced by a participant in the sketch production study was shown again to the same participant when they did the sketch recognition study. **Nevertheless, we agree that this detail should be included in the manuscript so we have added a sentence on L110-111 noting this**. As for the 1,481 participants who completed the machine sketch recognition study, here too there were 437 participants who also did the sketch production study, they were asked to recognize CLIPasso sketches and not human made sketches, so it was never the case that the same participant both produced and recognized the same sketch.

---

> > > ### Comment · Reviewer_p23D · 2023-08-30
> > > **RE: rebuttal**
> > >
> > > Thanks to the authors' rebuttal. I have read the rebuttal as well as the comments from other reviewers. I appreciate the authors' effort in addressing the concerns and providing additional evidence, which well address my previous concerns. Therefore I will keep my initial positive rating.
> > >
> > > There is still one (minor) suggestion for improvement, i.e. the actual time distribution (Supp.Fig.2), the authors used log response time instead of the actual time taken. It would be better to clarify the reason and include a description of this figure in the text.
> > > There is also some text cut-off for the horizontal axis in Fig. 3.

---

### Official Review · Reviewer_d6px · 2023-07-22
**New dataset of human sketches with different level of abstractions and a benchmark of human vs. machine sketch recognition and sketch generation**

**Rating:** 7
**Confidence:** 4

**Strengths:**

The dataset collected is very large and will be very useful for followup works
Sketch recognition benchmark is comprehensive and contains 17 different recognition methods

The evaluation and comparison of both recognition and generation abilities are thorough and contain multiple metrics. Author did a good job in highlighting where vision models and humans agree and where a clear difference is presence. I especially appreciate the discussion and graphs presented in Fig. 7.

Observations regarding the similarity and the gap between human sketch generation and recognition are interesting and pave the motivation for future algorithms that will better follow human perception.


**Additional Feedback:**

Overall I appreciate the task and the comprehensive evaluation of the paper, and hope the author will made the dataset interface more convenient to use

**Clarity:**

Paper is clear and easy to follow


**Correctness:**

All statements in the paper seem to be well supported.
For example the underlying assumption that drawing time governs abstraction, and equivalent to the number of strokes is exemplified in fig. 4.


**Documentation:**


The Dataset is currently available only through a zip download, which doesn’t let users/reader to get a high level impression of the sketches. Would be nice to build a website that enables to explore the dataset visually without downloading all of it.


**Ethics:**

No ethical concerns


**Limitations:**

I believe the Clipasso is trained to mimic sketches of professional artist, where the crowdsourcing sketches are of amateare artist, I wonder if the drawing process of professionals is different and will resemble clipasso more closely


**Opportunities For Improvement:**

The paper should include much more visualization of the dataset

Author should present and discuss the crowdsourcing platform in the supplementals (e.g. with screenshots).


**Relation To Prior Work:**

The paper nicely covers related works both from the machine learning aspect and visual cognition aspect, and also mentions existing sketch dataset (that do not include different abstraction levels). I wonder if the phenomena seen in the paper are related to “human classification”, if yes, it would be nice to add this to support the paper claims (e.g. https://www.nature.com/articles/s41467-020-18946-z).


**Summary And Contributions:**

The paper presents a new dataset of more than 80k human sketches with different levels of abstractions (different drawing durations) that were collected from more than 5K users. The paper suggests a benchmark that Measures the similarity between human and machine-learning systems in recognizing sketches as functions of abstraction level. The authors compare three scores: classification accuracy, classification entropy, and semantic uncertainty, and show that although general agreement in trends across time, learning algorithms still poses a gap to be human-like understanding. The paper also compares human and machine generated sketches with varying abstraction levels using Clipasso with different numbers of strokes, and shows that this resembles the human drawing process only to a limited extent.

---

> ### Author Response · Authors · 2023-08-21
> **Response to Reviewer d6px**
>
> We thank the reviewer for taking the time to read our paper and for the helpful comments.
> Below we address the concerns raised:
>
> > The paper should include much more visualization of the dataset
>
> Thank you for this suggestion! We agree that the paper would benefit from a figure that better captures the breadth of concepts represented and the variety of sketches in SEVA. **We have newly included Figure 2. on page 4, which showcases both human and machine sketches of 5 broad categories of concepts — animals, clothing, food, objects, and vehicles**.
> Additionally, if the paper were to be accepted, we plan to host an interactive dataset explorer on our project website to make it easier for other members of the research community to explore our dataset.
>
> > Author should present and discuss the crowdsourcing platform in the supplementals (e.g. with screenshots).
>
> **We have now included Sections S.3 and S.4 in the supplementary materials containing screenshots of both the sketching task and recognition task interface along with accompanying details of the tasks**. Information about participant recruitment on Prolific, compensation, and IRB are included in the main text under the Participants section of each study.
>
> > I wonder if the phenomena seen in the paper are related to “human classification”, if yes, it would be nice to add this to support the paper claims (e.g. https://www.nature.com/articles/s41467-020-18946-z).
>
> Thank you for pointing us to this very relevant reference! **We now include a reference to this paper (among others) on line 60**.

---

> > ### Comment · Reviewer_d6px · 2023-08-22
> > **Thank you for the response**
> >
> > I appreciate the addition of fig. 2 and the additional references.
> > The author addressed all my concerns, I keep my rating the same.

---

### Official Review · Reviewer_Jrh3 · 2023-07-25
**A useful dataset and valuable experiments**

**Rating:** 7
**Confidence:** 4
**Correctness:** Yes.
**Clarity:** Yes, it is easy to follow.

**Strengths:**

- The dataset is of large-scale.
- Sketch understanding especially for different-level of abstractions is also a very important topic.


**Additional Feedback:**

No.

**Documentation:**

Yes.

**Ethics:**

NA.

**Limitations:**

NA.

**Opportunities For Improvement:**

- Regarding dataset, I suggest the authors to include more discussions on the differences with existing dataset. A table is highly encouraged to show the comparisons.


**Relation To Prior Work:**

See above.

**Summary And Contributions:**

The work presented a new dataset of human-level sketches, which covers 128 object concepts. For each sketch, it contains different levels of abstraction. Overall, I think the dataset is valuable and I also very appreciate the authors' efforts to provide many experiments and analysis.

---

> ### Author Response · Authors · 2023-08-21
> **Response to Reviewer Jrh3**
>
> We thank the reviewer for taking the time to read our paper and for the helpful comments. We are glad to hear that the reviewer sees the value in the dataset we developed, as well as the experiments and analyses we conducted.
>
> Thank you for the terrific suggestion:
>
> >Regarding dataset, I suggest the authors to include more discussions on the differences with existing dataset. A table is highly encouraged to show the comparisons.
>
> **In our revision, we now include a new table (Table 1,  page 4) and new text (line 89) that explicitly notes similarities and differences between SEVA (our dataset) and other existing sketch datasets**. We hope that providing this additional information better contextualizes our contribution.

---

### Decision · Program_Chairs · 2023-09-22

**Decision:**

Accept (Poster)

**Comment:**

The paper introduces a new dataset of human sketches with varying levels of abstraction and proposes a benchmark study comparing human and machine visual abstraction in sketch recognition. The dataset is of large-scale and contains valuable information for follow-up research. The evaluation is comprehensive, with multiple recognition methods and metrics used. The findings are interesting and relevant to better understanding human perception and guiding future algorithms. However, there are opportunities for improvement, including providing more visualizations of the dataset and addressing concerns raised by reviewers regarding the dataset collection process, the need for human-like models, and potential overlaps in participant recruitment. The limitations and ethical considerations also need to be addressed, and additional experiments and comparisons, such as with recurrent models, could be considered.